# Effects of Rhenium Substitution of Co and Fe in Spinel CoFe_2_O_4_ Ferrite Nanomaterials

**DOI:** 10.3390/nano12162839

**Published:** 2022-08-18

**Authors:** Yuruo Zheng, Ghulam Hussain, Shuyi Li, Shanta Batool, Xiawa Wang

**Affiliations:** 1Department of Natural and Applied Sciences Duke Kunshan University, Suzhou 215316, China; 2Department of Physics, and Key Laboratory of Strongly-Coupled Quantum Matter Physics (CAS), University of Science and Technology of China, Hefei 230026, China

**Keywords:** sol-gel method, cobalt ferrite, substitution, vibrational modes, magnetization, dielectric properties

## Abstract

In this work, nanoparticles of Co_1−x_Re_x_Fe_2_O_4_ and CoFe_2−x_Re_x_O_4_ (0 ≤ x ≤ 0.05) were synthesized by the sol-gel method. The Rietveld refinement analysis of XRD and Raman data revealed that all of the prepared samples were single phase with a cubic spinel-type structure. With the substitution of Re, the lattice parameters were slightly increased, and Raman spectra peak positions corresponding to the movement of the tetrahedral sublattice shifted to a higher energy position. Furthermore, Raman spectra showed the splitting of T_2g_ mode into branches, indicating the presence of different cations at crystallographic A- and B-sites. The SEM micrograph confirms that surface Re exchange changes the coordination environment of metals and induces Fe-site structure distortion, thereby revealing more active sites for reactions and indicating the bulk sample’s porous and agglomerated morphology. The vibrating sample magnetometer (VSM) results demonstrated that the synthesized nanoparticles of all samples were ferromagnetic across the entire temperature range of 300–4 K. The estimated magnetic parameters, such as the saturation magnetization, remanent magnetization, coercivity, blocking temperature (*T_B_*), and magnetic anisotropy, were found to reduce for the Co-site doping with the increasing doping ratio of Re, while in the Fe site, they enhanced with the increasing doping ratio. The ZFC-FC magnetization curve revealed the presence of spin-glass-like behavior due to the strong dipole–dipole interactions in these ferrite nanoparticles over the whole temperature range. Finally, the dielectric constant (*ε_r_′*) and dielectric loss (*tan**δ*) were sharply enhanced at low frequencies, while the AC conductivity increased at high frequencies. The sharp increases at high temperatures are explained by enhancing the barrier for charge mobility at grain boundaries, suggesting that samples were highly resistive. Interestingly, these parameters (*ε_r_′*, *tan**δ*) were found to be higher for the Fe-site doping with the increasing Re doping ratio compared with the Co site.

## 1. Introduction

Spinel ferrite has wide applications in electronic devices, environmental protection, and biomedical technology [1,2,3,4]. In recent years, substituted spinel ferrite has attracted considerable attention from numerous researchers thanks to remarkable properties such as reliable saturation magnetization (*Ms*) [5], high anisotropy (*K*) [6,7], high coercivity (*H_c_*) [8], high Curie temperature (*T_c_*) [9], thermal stability, magneto-optical properties, catalytic, biocompatibility, high electrical resistivity [2,10,11], and so on. Spinel ferrite refers to the Fd3m cubic spinel structure with a unit formula of AFe_2_O_4_, where A represents divalent and Fe^3+^ represents trivalent metal elements. The unit cell of AFe_2_O_4_ consists of 32 closely packed oxygen atoms with 64 divalent tetrahedral A-sites and 32 trivalent octahedral B-sites. Only 8 of the available 64 A-sites are occupied by cations, and only 16 of the 32 B-sites are occupied by cations. Furthermore, spinel ferrites have received significant attention because of metal ion substitution and anion exchange phenomena attributes to the enormous electronic and geometric structure, which usually leads to charge separation, carrier migration, and variation in the energy bands. The metal ion-dependent physical properties are mostly inspired by the synthesis approach and the cation distribution between the tetrahedral and octahedral sites. Recently, it has been extensively reported that the structural defect, particle size, shape, composition, and arrangement of A^2+^ and Fe^3+^ cations can strongly affect the crystal structure and interactions between ions, altering spinel ferrites’ magnetic, electrical, and optical properties [1,12,13,14,15]. Therefore, controlled substitution and site preference (A-sites or trivalent octahedral B-sites) can improve spinel ferrites’ properties [16,17].

Among spinel ferrite materials, cobalt ferrites (CoFe_2_O_4_) are used in various applications because of their good electrical insulation, high coercivity, and easy synthesis method [18,19,20]. Mostly, these cobalt ferrite nanoparticles are often synthesized by different techniques such as solid-state reaction [21], ultrasound irradiation [22], ball milling [19], sol-gel auto combustion [23] methods, and so on. Among these techniques, the sol-gel approach is the most appropriate for synthesizing nanoparticles with controlled sizes, high homogeneity, and proper microstructure. There have been many previous efforts to study these materials. Koseoglu et al. synthesized Mn-doped cobalt ferrite and discussed the variation in magnetic properties and the potential application in magnetic sensing [24]. Muscas et al. substituted Co^2+^ with zinc and discovered an increasing coercivity behavior as a result of anisotropy change [25]. Kavitha et al. used zirconium to replace cobalt ions, reporting improved electrical conductivity, dielectric loss, and coercivity field [26]. The search for new dopants improves the cobalt ferrite’s structure and physical properties. Rhenium belongs to the 5d^3^ transition metal, which has been reported as a superior dopant of some semiconductors and oxides to change the properties of the hosting materials. Teli et al. discussed the potential spintronic and optoelectronic application of Rhenium-doped alkaline earth oxides through density functional theory (DFT) calculation [27]. Very recently, Assadi et al. predicted through theoretical calculation that ReFe_2_O_4_ occupies an unconventional ferrimagnetic state with potential application in spintronics [28]. Moreover, Heidari et al. reported the superior thermal plasmonic characteristics of rhenium nanoparticles and their potential application in tumor treatment [29]. In this work, we synthesized rhenium (Re)-doped cobalt ferrite nanoparticles. We performed XRD, SEM, Raman, M-H, ZFC-FC, and dielectric tests to investigate its influence on cobalt ferrite’s structural, optical, and magnetic properties. We substituted Re atoms in different sites that can alter the ferrite’s magnetic moment and crystal structure, resulting in changes in the properties of the hosting material. 

## 2. Experiment

### 2.1. Preparation of Co_1−x_Re_x_Fe_2_O_4_ and CoFe_2−x_Re_x_O_4_ Nanoparticles

The nanoparticles of Co_1−x_Re_x_Fe_2_O_4_ and CoFe_2−x_Re_x_O_4_ (0 ≤ x ≤ 0.05) were synthesized by the sol-gel method. Both series of nanoparticles undergo the same procedure with different stoichiometric ratios. During the synthesis, rhenium powder (metals basis, 99.99%, Aladdin, Shanghai), Co(NO_3_)_2_·6H_2_O (99%, damas-beta), and Fe(NO_3_)_3_·9H_2_O (98%, damas-beta) were used as oxidizing agents. The ingredients were weighed in desired stoichiometric proportions, and the metal nitrates were mixed and dissolved in 150 mL of distilled water using a magnetic stirrer at room temperature. Rhenium was dissolved in deionized (DI) water and placed in an ultrasonic bath at 40 °C for 30 min. The base solution of metal nitrates and rhenium solution were mixed and stirred for 1 h at 30 °C. Then, 50 mL 0.1 M HNO_3_ was added to the mixture and stirred for 30 min. Afterwards, 80 mmol ethylene glycol (EG, C_2_H_6_O_2_) was added to foster the formation of sol and prevent agglomeration. The precursor’s mixture was heated to 80 °C and stirred steadily on a hot plate until a puffy dark brown gel appeared. The gel was then transferred into a dish for combustion and maintained at the same temperature. This study investigates the effects of Rhenium substitution for Co and Fe on the structural and physical properties of inverse-spinel CoFe_2_O_4_.

The viscous gel began to froth until a dried gel was achieved. Finally, the dry gel was calcinated at 800 °C for 3 h to obtain crystalline cobalt ferrite nanoparticles. The targeted nanoparticle sample was obtained by grinding the bulk with a mortar and pestle for 40 min. For further clarification, the synthesized process of the required rhenium-doped cobalt ferrite is illustrated in Figure 1.

### 2.2. Characterization Method

Phase identification and impurity detection were carried out with an X-ray diffraction (XRD) method using an Aeris model (Aeris, Malvern Panalytical, Malvern, UK) with Cu kα radiation with 2θ from 10° to 80°. SEM (Regulus 8100, Hitachi, Tokyo, Japan) was used to investigate the morphology of the samples. A vibrating sample magnetometer (PPMS-9T, Quantum Design, San Diego, CA, USA) was used to obtain magnetic hysteresis loops and ZFC-FC curves at room and low temperatures. Moreover, Raman spectra were measured by confocal Raman microscope (LabRAM HR Evolution, Horiba, Kyoto, Japan). Dielectric constant and dielectric loss were obtained by a broadband dielectric/impedance analyzer (Concept 80, Novocontrol, Montabaur, Germany).

## 3. Results and Discussion

### 3.1. XRD Analysis

We performed the X-ray diffraction (XRD) tests of the sol-gel prepared samples to investigate the phase purity and crystal structure of the acquired (Co_1−x_Re_x_Fe_2_O_4_ and CoFe_2−x_Re_x_O_4_) nanoparticles. All samples were tested in continuous mode from 10° to 80°, indicating the successful formation of the single phase without any distortion. Figure 2 shows the XRD spectra of the Co_1−x_Re_x_Fe_2_O_4_ and CoFe_2−x_Re_x_O_4_ (x = 0, 0.02, 0.05) prepared samples. After further refinement by the software Fullprof, all XRD peaks were well-indexed. It can be seen that all samples have prominent peaks that correspond to (111), (220), (311), (400), (422), and (440) planes, which all belong to the Fd3m cubic spinel structure, indicating that Re does not change the structure of the hosting material and maintains a pure single phase of cobalt ferrite nanoparticles. Furthermore, the experimental XRD data were used with the software JANA 2006 to estimate the lattice parameters, surface-to-volume ratio, and reliability factors. The average crystallite size was calculated from the linewidth of (311) broadening of the XRD pattern with the Scherer formula [12]:(1)$$D=\frac{0.9\lambda }{\beta cos\theta }$$
where *D* shows the crystallite size in nm, *θ* represents the diffracted Braggs angle, *β* defines the full width of half maximum of the diffraction peaks, and λ is the wavelength of Cu-Kα. The lattice constant (a) is determined by the following equation:
(2)
1/d^2^ = (h ^2^ +k^2^ +l^2^)/a^2^


h, k, and l are Miller indices, and d is the interplanar spacing. The values of the lattice constant (a = b = c(Å)) increased slightly with the doping ratio, indicating the strong presence of Re ion in our samples. The refined structural parameters of pure and Re-doped cobalt ferrite nanoparticles are presented in Table 1. Reliability factors such as goodness of fitting (GOF), weighted profile R-factor (R_wp_), and reliability factor (R_p_) of Rietveld refinement were also evaluated.

Firstly, the grain size decreases slightly with the increasing amount of doping, possibly resulting from the strain effect of Co and Re ions. The structural strain prevents the growth of crystallites by the substituting elements. With the small amount of doping, the doping of Re induced a lattice shrinkage of the neighbor lattices, causing a reduction in the grain size. Secondly, the Re^4+^ radius in the octahedral site is 0.63 Å, which is rather close to the replaced Fe^3+^ radius (0.65 Å) and is not expected to be a substantial driver of the structural transformation [28]. The inset of Figure 2 shows that the lattice constant increases with the Re doping ratio on both Co and Fe sites. This increase may be due to carrier hopping between the dopants and the hosting lattice, as the doping of Re^4+^ will change the Fe ions from trivalent to divalent [28]. Many other heavy doping cobalt ferrites show a similar lattice shrinkage in the literature. For example, Nongjai and Satheeshkumar et al. synthesized indium- and silver-doped cobalt ferrite nanoparticles and reported an expansion in lattice parameters due to the substitution of large radius ions [14].

### 3.2. SEM and EDS Analysis

The surface morphology information of the synthesized Co_1−x_Re_x_Fe_2_O_4_ and CoFe_2−x_Re_x_O_4_ (0 ≤ x ≤ 0.05) nanoparticles was inspected by scanning electron microscopy (SEM). The pure and Re-doped cobalt ferrite SEM images were taken at a 5 μm scale, as shown in Figure 3a–e. The SEM results indicate the formation of nanoparticles with an inhomogeneous spherical or semi-spherical strain distribution. Nonuniform strain distribution can cause a variation in the size and shape of the particles. The nanoparticles’ formation was composed of multi-particle agglomerations, which can be seen clearly for the sample Co_1−x_Re_x_Fe_2_O_4,_ which increases with the content of Re in the Co site. In the literature, many previous works have reported the agglomerations. For example, Mritunjoy et al. observed similar agglomeration behavior when doping the cobalt ferrite with Cu, asserting that Van der Waals force and dipole–dipole interaction mainly contributed to this enhancement in the interparticle interactions [9]. Shakil et al. also reported the agglomeration of Zn and Cd co-doped cobalt ferrite, proposing that the interactions of magnetic moments between different lattice sites were likely to result in this agglomeration [30]. When doping cobalt ferrite with Ti, the grain size increases with a higher doping ratio, which can be attributed to the interactions between magnetic particles [31].

Furthermore, it is noted that, with the increase in the Re doping ratio, the Co-site doped sample particles agglomerate faster than the Fe-site doped samples, which may be due to the dipole–dipole interactions between each grain. The agglomeration is severely reduced by adding Re to the Fe site, as seen in Figure 3e,f. Two different morphologies can be seen with Re Co-site and Re Fe-site doping, thus larger particles can be associated with the existence of interparticle interactions.

Figure 4 compares the grain size of the Co_1−x_Re_x_Fe_2_O_4_ and CoFe_2−x_Re_x_O_4_ samples calculated from the XRD and SEM result. It can be seen that the particle size measured from the SEM image is more than 10 times larger than the grain size calculated by XRD spectra, further indicating an agglomeration behavior of the particles, which has also been reported by other studies [32,33].

### 3.3. Raman Analysis

Raman spectroscopy is a powerful technique to investigate the structure of nanoparticles, especially for tracing the spinel’s structural evolution and cation distribution [34,35,36]. The room temperature Raman spectra of (a) CoFe_2_O_4_, (c) Co_0.98_Re_0.02_Fe_2_O_4_, (d) Co_0.95_Re_0.05_Fe_2_O_4_, (e) CoFe_1.98_Re_0.02_O_4_, and (f) CoFe_1.95_Re_0.05_O_4_ nanoparticles are presented in Figure 5a–f, and the spectra were further deconvoluted into seven individual Lorentzian peaks.

According to the XRD analysis, Re-doped cobalt ferrite belongs to the cubic space group Fd3m, which has five Raman active modes (A_1g_: 648–680 cm^−1^; E_g_: 278–293 cm^−1^; and 3T_2g_: 539–565, 449–471, and 163–177 cm^−1^), predicted from the group theory analysis [37,38,39]. It is seen from Figure 5a that the pure cobalt ferrite sample shows five major bands at ~195, 300, 462, 562, and 684 cm^−1^, which correspond to the five predicted Raman active modes (A_1g_ + E_g_ + 3T_2g_), confirming the formation of spinel phases. Besides, two bands at ~621 cm^−1^ and ~356 cm^−1^ also appear as a result of the quantum size effect and the inverse and mixed-type spinel ferrite reported in the literature [40,41,42]. The peak intensity and position depend on the vibrational modes and the distribution of cations. Specifically, A_1g_(2) and A_1g_(1) peaks at 621 and 685 cm^−1^, respectively, can be attributed to the Fe^3+^-O and Re^2+^-O bonds at tetrahedral voids (AO_4_) [40,43]. The T_2g_(2) and T_2g_(1) at 460 and 567 cm^−1^, respectively, are attributed to the asymmetry bending motions of the oxygen in Fe^3+^-O and Re^2+^-O bonds at tetrahedral voids, and T_2g_(3) at 195 cm^−1^ is derived from the vibration of cations in octahedral voids [44]. The E_g_ peaks at 300 cm^−1^ correspond to the symmetric oxygen bending in AO_4_ units [45,46].

Figure 5b demonstrates the evolution of Raman peak position and peak intensity with respect to the Re doping ratio. It can be seen that, with the increase in the doping ratio, the peak positions of the A_1g_ (685 cm^−1^), T_2g_(2) (460 cm^−1^), and E_g_ (300 cm^−1^) modes increase. According to the literature, these three peaks originate from the movement of the tetrahedral voids, which indicates that the inclusion of Re mainly influences the structure of the tetrahedral sites, i.e., increasing the vibrational energy of Fe^3+^–O and Re^2+^–O bonds by varying the bond length [41]. Regarding the T_2g_(3) (195 cm^−1^) mode, the peak position increases with the Fe-site doping ratio, while its fluctuates slightly for Co-site doping samples. This implies that the substitution of iron to Re also influences the structure of octahedral sites, leading to a more distorted structure compared with Co-site doping. Other modes’ peak positions are altered slightly with the increase in the doping ratio. The shift in Raman peaks indicates that Re substitution imposes a distortion of the tetrahedral sublattice and influences the octahedral sublattice for Fe-site doping.

### 3.4. Magnetic Analysis

The magnetic hysteresis loop (M-H) tests of the pure and Re-doped cobalt ferrite at 300 and 4 K are illustrated in Figure 6a,b. The M-H curve was recorded in a magnetic field up to 50 kOe at temperatures of 300 and 4 K. However, at temperatures of 300 and 4 K, our M-H data show the single phase of the ferromagnetism. Based on the analysis of all these hysteresis loops, we extracted the saturation magnetization, coercivity, and remanence magnetization. It is observed that all of the pure and Re-doped cobalt ferrite nanoparticles show hard ferromagnetic behavior, as expected. These measured values of M-H of the samples were consistent with the reported values and were strongly temperature-dependent [47]. However, the magnetic properties can be improved by substituting metal ions, which usually leads to charge separation, carrier migration, and variation in energy band structure used to control the electronic structure of the spinels. The extracted parameters from the M-H loops, such as saturation magnetization (*M_s_*), coercivity (*H*_c_), magnetic anisotropy (*K*), remanence magnetization (*M_r_*), squareness ratio (*Mr/Ms*), and magnetic moment (*n_B_* in Bohr magneton), are listed in Table 2. Moreover, the anisotropy constant and Bohr magneton are calculated from Equations (3) and (4) [48]:(3)$$K=\frac{H_{c}\times M_{s}}{0.98}$$
(4)$$n_{B}=\frac{M_{CoFe2O4}\times M_{s}}{5585}$$
where *H*_c_ is the coercivity, *M*_s_ is the saturation magnetization (emu/g), and *M_CoFe_*_2_*_O_*_4_ is the molecular weight of the ferrite. For the Co-site doping, the saturation magnetization and remanence magnetization decreases with the increasing doping ratio, while in the Fe-site, they are enhanced with the increasing doping ratio. For the Co-site doping, the saturation magnetization decreases with the increase in the doping ratio from 0 to 0.05 (at 300 K, from 80 to 78 emu/g; at 4 K, 80 to 77 emu/g), while for the Fe-site doping, the saturation magnetization increases (from 80 to 81 emu/g at 300 K; 80 to 82.6 emu/g at 4 K). The variation in saturation magnetization is closely related to the size and shape of the nanoparticles. From SEM, it is observed that the particles agglomerate with the increase in substitution of Re to Co as a result of an increase in dipole interactions, changing the surface properties of the particles. It is likely that, because of agglomeration resulting from Re doping, the surface-to-volume ratio of the particle decreases, reducing the surface effect, and the orientation of the magnetic moment can also be influenced by the neighboring particles. However, the particle size decreases slightly with the increase in doping of Re to Fe, increasing the saturation magnetization. Additionally, the coercivity measured at 4 K decreases with the increase in the Re–Co substitution ratio, while it increases with the increase in doping of Re to Fe. This can be explained by the change in anisotropy with the inclusion of Re. Moreover, the magnetic moment increases with the decrease in the Co-site doping ratio, which can be explained by the change in magnetic anisotropy. The magnetic anisotropy decreases for Co-site doping, leading to a more random orientation of the moment and a decrease in the net magnetic moment. Similarly, the anisotropy of Fe-site doping increases, resulting in an increase in the total magnetic moment with an increase in the doping ratio. Compared with other dopants, Re-doped cobalt ferrite maintains a higher total magnetic moment [44,49], indicating an increase in the anisotropy resulting from doping Re.

When comparing Co-site doping and Fe-site doping, the decrease in *M_s_* and *H_c_* in Co-site doping and increase in *M_s_* and *H_c_* in Fe-site doping are closely related to the change in magnetocrystalline anisotropy resulting from Re doping. As shown in Figure 6c, the anisotropy at 4 K decreases in Co-site doping and increases in Fe-site doping. Furthermore, this anisotropy behavior can be attributed to the difference in the total energy of the structures [29,48].

When Re is doped in the Co site, Re mainly occupies octahedral sites with opposite spin alignment with iron in both the octahedral and tetrahedral sites. This structure appears to be at the lowest energy state and results in a decrease in anisotropy. When doping Re in the Fe site, the structure occupies a higher energy state, increasing anisotropy as a result of replacing heavy Re ions with smaller Co^2+^ ions at the octahedral sites. This decreasing and increasing behavior mainly results from the surface effects generated from the grain size variation. Moreover, it is observed that, at a low temperature (4 K), the remanence magnetization, coercivity field, and squareness of all samples are higher than those at 300 K. Figure 6c shows the variation in anisotropy with the doping ratio measured at 4 K and 300 K. It is observed that the anisotropy increases significantly with the doping of Re in the Fe site. In contrast, in the Co site, the anisotropy decreases. It is also worth noting that the hysteresis loop of the Co-site and Fe-site doping show different shapes. Especially at 4 K, Fe-site-doped samples show a higher coercivity and saturation magnetization, indicating better magnetic performance than Co-site doping. The significant increase in coercivity mainly results from the increased magnetic anisotropy from substituting iron with rhenium. Magnetization versus temperature characterizations were also conducted on different samples. Figure 6d presents the ZFC-FC magnetization curve measured under 50 kOe of Co_1−x_Re_x_Fe_2_O_4_ and CoFe_2−x_Re_x_O_4_ nanoparticles, showing an irreversible behavior. It can be seen from the figure that the blocking temperature of all of the nanoparticles is about 250 K, with a slight increase in Fe-site doping and a decrease in Co-site doping, which reflects the variation trend of anisotropy at 4 K.

### 3.5. Dielectric Behavior

The dielectric properties of the spinel ferrite are strongly determined by the chemical composition and site occupancy of metal cations [50]. To further investigate the influence of different site doping, the dielectric properties of CoFe_2_O_4_, Co_0.98_Re_0.02_Fe_2_O_4_, and CoFe_1.98_Re_0.02_O_4_ nanoparticles were obtained in various temperature and frequency ranges of 1 Hz to 1 MHz with an applied AC electric field. The obtained results of the dielectric constant, tangent loss, and AC conductivity are shown in Figure 7, Figure 8 and Figure 9. The following equations are used to calculate the dielectric constant, dissipation energy, and AC conductivity:(5)$$\;\varepsilon_{r}^{\prime }=\frac{Cd}{A\varepsilon_{o}}$$
(6)$$tan\delta =\frac{\varepsilon^{\prime \prime }}{\varepsilon_{r}^{\prime }}$$
(7)$$\sigma_{\mathrm{ac}}=\omega \varepsilon_{0}{\varepsilon^{\prime }}_{r}tan\delta$$

Figure 7a–c show the real part (*ε_r_’*) of the dielectric constant versus frequency at various temperatures for CoFe_2_O_4_, Co_0.98_Re_0.02_Fe_2_O_4_, and CoFe_1.98_Re_0.02_O_4_ nanoparticles. The insets of Figure 7a–c show the temperature-dependent real part of each plot. The real part of the dielectric dispersion was found to have the highest value at a low frequency and decreases at a high frequency at all temperatures (CoFe_2_O_4_, Co_0.98_Re_0.02_Fe_2_O_4_, and CoFe_1.98_Re_0.02_O_4_), representing the typical behavior of spinel ferrite nanoparticles. The real part undergoes a sharp decrease with frequency up to 1 kHz and remains constant at a high frequency. Such behavior could be described by considering many sources of dielectric polarization, such as ionic or interfacial polarization. At lower frequencies, the main contribution to the polarization is ionic, which is thus responsible for the high value of the real part. In comparison, at higher frequencies, only electronic polarization remains active because of the inability of electric dipoles to follow the fast variation in the alternating electric field. The contribution of net ionic and orientation polarizability becomes slow at higher frequencies, which causes a decrease in the real part and finally remains almost constant. The decrease in ionic polarizability or initial loss can be attributed to orientation.

The tangent loss factor ($tan\delta =\frac{\varepsilon^{\prime \prime }}{\varepsilon_{r}^{\prime }}$) is defined as the ratio of the dissipated energy to the stored energy in the material. The values of tanδ as a function of frequency at various temperatures of CoFe_2_O_4_, Co_0.98_Re_0.02_Fe_2_O_4_, and CoFe_1.98_Re_0.02_O_4_ nanoparticles are shown in Figure 8a–c. The temperature variation is also shown in the insets in Figure 8a–c. It is observed that tanδ displays the highest value at a low frequency and increases with temperature. The highest value of *tanδ* in the low-frequency region can be attributed to the high resistivity of grain boundaries, which are more effective at lower frequencies. The energy loss may be attributed to collision, vibration, and other charge particle interaction phenomena. The AC conductivity versus frequency at various temperatures of CoFe_2_O_4_, Co_0.98_Re_0.02_Fe_2_O_4_, and CoFe_1.98_Re_0.02_O_4_ nanoparticles is shown in Figure 9a–c. The values of AC conductivity were almost independent for all samples at a low frequency and displayed polarization at a high frequency. At a low frequency, the hopping of charges across the grain boundaries is minimal and increases with the frequency. The variation in temperature-dependent AC conductivity is shown in the inset of each plot in Figure 9a–c. It is evident that the dielectric constant decreases slightly when doping Re at the Co-site, while it increases significantly in Fe-site doping. The Re doping in the Fe site is expected to induce the formation of small polarons, while reducing them when doping in the Co-site. The polarization in cobalt ferrites is mainly attributed to electron hopping between Fe^2+^ and Fe^3+^ ions and hole hopping between Co^3+^ and Co^2+^ ions [7,51,52]. With Re substituting Co, the total number of hole sources decreases, which reduces the accumulation of the carrier, thus decreasing the polarization. For Fe-site doping, the substituion of Fe^3+^ by Re^4+^ induces electron hopping, which supports the formation of polarons. Moreover, Rahman et al. reported that the formation of polaron derives from the increase in structure distortion, which means that Fe-site doping increases the structure distortion, while Co-site doping decreases the distortion [53], confirming the variation in anisotropy discussed in the magnetic parts.

## 4. Conclusions

In summary, the effect of Rhenium Co-site and Fe-site doping on the structural, microstructural, magnetic, Raman, and dielectric properties of CoFe_2_O_4_ synthesized by the sol-gel approach was explored. XRD Rietveld refinement and Raman results indicated that all of the synthesized samples were single phase with cubic spinel-type structures, and the lattice parameters increased slightly with the substitution of Rhenium. In the Raman data, we find that, with an increased doping ratio, most of the peak positions that correspond to the movement of the tetrahedral sublattice shift to a higher energy site, and the doping of Re induces the distortion of the crystal structure. The SEM micrograph confirms that the surface Re exchange changes the coordination environment of metals and induces Fe-site structure distortion, thereby revealing more active sites for reactions. Furthermore, the calculated magnetic parameters, such as saturation magnetization, remanent magnetization, coercivity, blocking temperature (*T_B_*), and magnetic anisotropy, decreased in the Co-site doping with the increasing doping ratio of Re, while in the Fe site, they were enhanced with the increasing doping ratio. The variation in saturation magnetization is closely related to the size and shape of the nanoparticles. From SEM, it is observed that the particles agglomerate with the increase in the substitution of Re to Co owing to an increase in dipole interactions, changing the surface properties of the particles. It is likely that, because of agglomeration resulting from Re doping, the surface-to-volume ratio of the particle decreases, reducing the surface effect, and the neighboring particles can also influence the orientation of the magnetic moment. Finally, the dielectric measurement revealed that the real part (*ε_r_’*) and tangent loss (*tan**δ*) show a response at low frequencies. The sharp increase at high temperatures enhances the barrier for charge mobility at grain boundaries, suggesting that samples were highly resistive. The real part and the tangent loss were found to be higher in the Fe-site doping with the increasing Re doping ratio as compared with the Co site, which was attributed to interfacial or space charge polarization.

## Figures and Tables

**Figure 1 nanomaterials-12-02839-f001:**
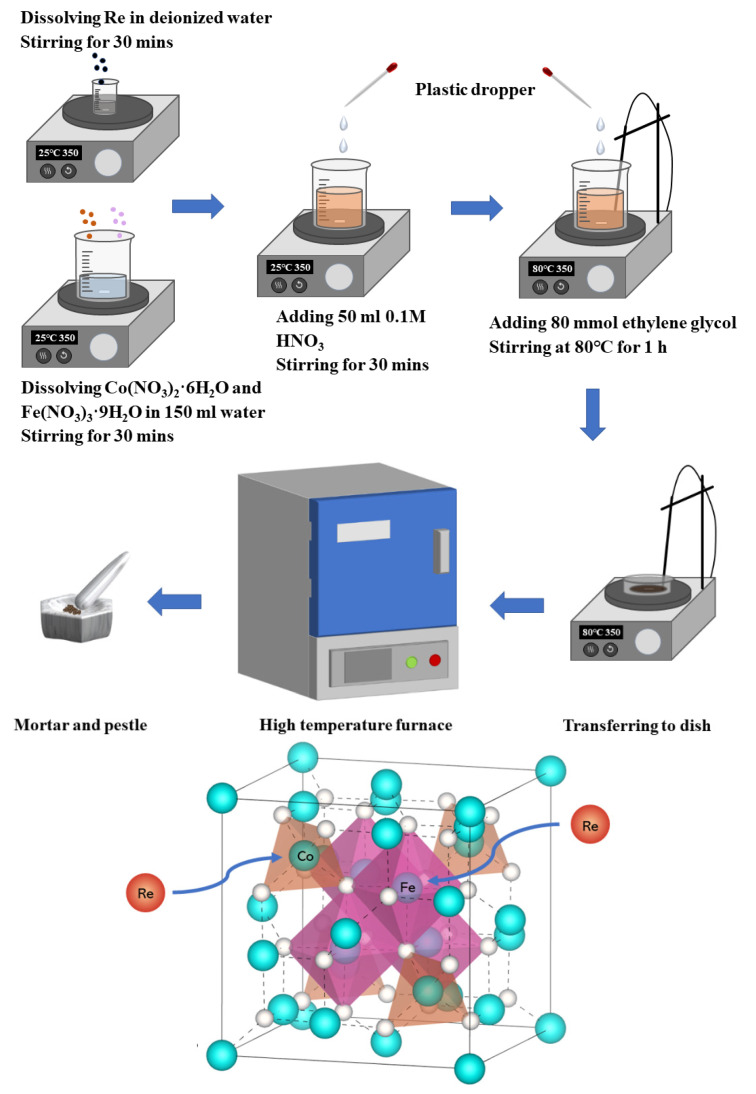
The crystal structure of the rhenium-doped cobalt ferrite illustrates the sol-gel synthesis procedure.

**Figure 2 nanomaterials-12-02839-f002:**
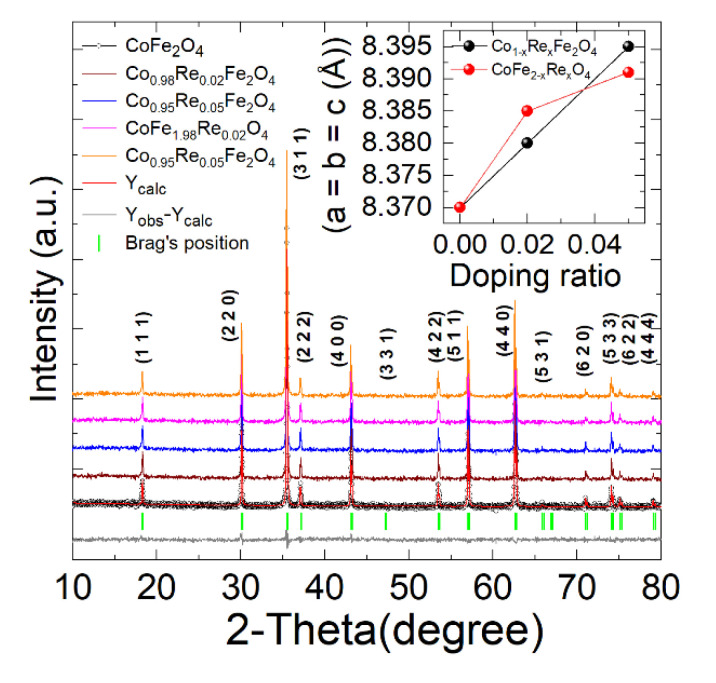
XRD powder pattern of CoFe_2_O_4_, Co_0.98_Re_0.02_Fe_2_O_4_, Co_0.95_Re_0.05_Fe_2_O_4_, CoFe_1.98_Re_0.02_O_4_, and CoFe_1.95_Re_0.05_O_4_ nanoparticles. All samples show a pure Fd3m cubic spinel structure. The inserted figure shows the refined cell parameters’ variation with respect to the doping ratio; the cell parameter increases with the increase in the rhenium doping ratio.

**Figure 3 nanomaterials-12-02839-f003:**
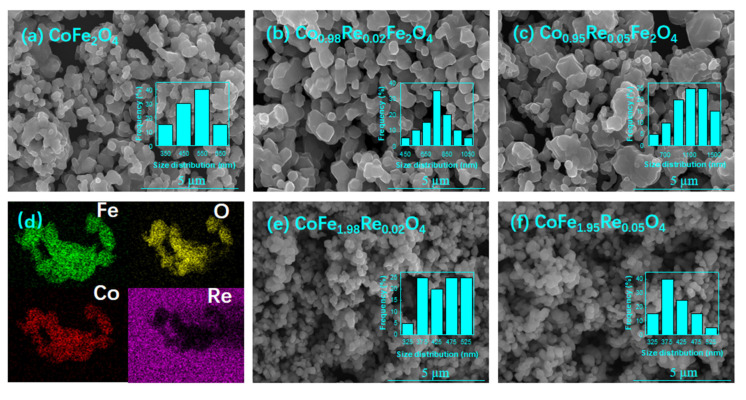
SEM image of (**a**) CoFe_2_O_4_, (**b**) Co_1.98_Re_0.02_Fe_2_O_4_, and (**c**) Co_1.95_Re_0.05_Fe_2_O_4_; (**d**) EDS mapping of Fe, O, Co, and Re elements for the Co_1.98_Re_0.02_Fe_2_O_4_ nanoparticle, showing a uniform distribution of the elements. SEM image of (**e**) CoFe_1.98_Re_0.02_O_4_ and (**f**) CoFe_1.95_Re_0.05_O_4_ nanoparticles. Agglomeration appeared for all doped samples. However, the agglomeration is severely reduced for Fe-site doped samples (**e**,**f**).

**Figure 4 nanomaterials-12-02839-f004:**
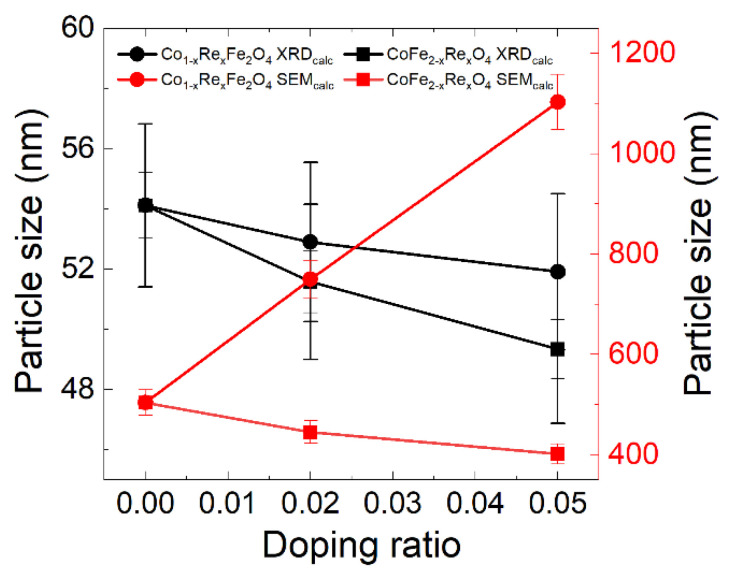
Grain size of the Co_1−x_Re_x_Fe_2_O_4_ and CoFe_2−x_Re_x_O_4_ samples calculated from the XRD and SEM result. The particle size is more than 10 times larger than the grain size.

**Figure 5 nanomaterials-12-02839-f005:**
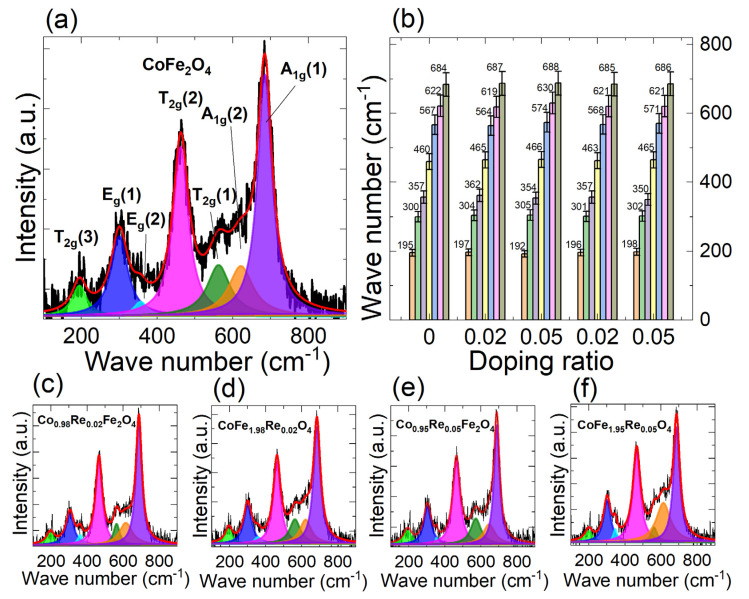
Room temperature Raman spectra and the fitting result of (**a**) CoFe_2_O_4_, (**c**) Co_0.98_Re_0.02_Fe_2_O_4_, (**d**) Co_0.95_Re_0.05_Fe_2_O_4_, (**e**) CoFe_1.98_Re_0.02_O_4_, and (**f**) CoFe_1.95_Re_0.05_O_4_ nanoparticles excited by a 0.17 mW 532 nm laser. (**b**) Evolution of the fitting peak intensity and position with the Re doping ratio. The spectra verify the inverse spinel phase, and the peak positions of the A_1g_ (685 cm^−1^), T_2g_(2) (460 cm^−1^), and E_g_ (300 cm^−1^) modes increase with Re doping, while the positions fluctuate for other modes, indicating a different presence of cations.

**Figure 6 nanomaterials-12-02839-f006:**
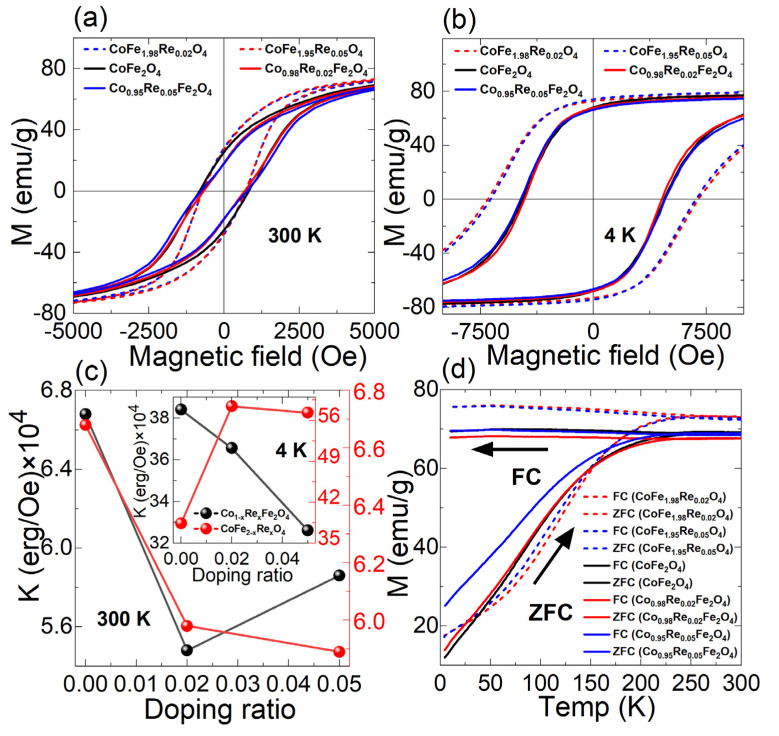
Magnetic hysteresis loops of Co_1-x_Re_x_Fe_2_O_4_ and CoFe_2-x_Re_x_O_4_ nanoparticles at (**a**) 300 K and (**b**) 4 K, showing a variation in the hysteresis shape with different sites of doping. (**c**) Variation in anisotropy at 300 K and 4 K with respect to the doping ratio. The anisotropy decreases for Co-site doping, while it increases for Fe-site doping. (**d**) M-T curve of Co_1−x_Re_x_Fe_2_O_4_ and CoFe_2−x_Re_x_O_4_ nanoparticles, indicating a spin-glass-like behavior of the nanoparticles.

**Figure 7 nanomaterials-12-02839-f007:**
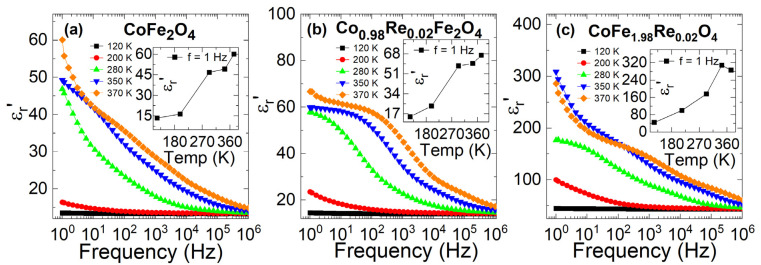
Variation in the dielectric constant of (**a**) CoFe_2_O_4_, (**b**) Co_0.98_Re_0.02_Fe_2_O_4_, and (**c**) CoFe_1.98_Re_0.02_O_4_ nanoparticles with different temperatures. The dielectric constant decreases significantly at high frequencies. The dielectric constant increases in Fe-site doping, while it decreases in Co-site doping.

**Figure 8 nanomaterials-12-02839-f008:**
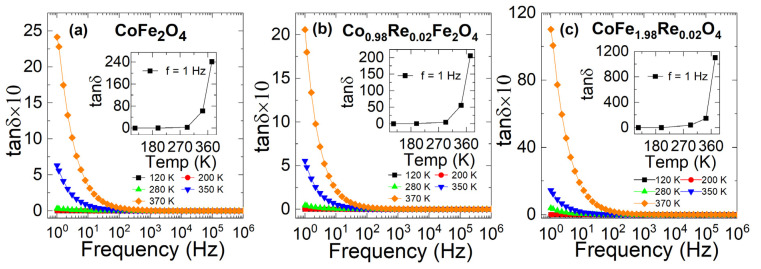
Variation in the energy dissipation of (**a**) CoFe_2_O_4_, (**b**) Co_0.98_Re_0.02_Fe_2_O_4_, and (**c**) CoFe_1.98_Re_0.02_O_4_ nanoparticles with different temperatures. The energy dissipation decreases at high frequencies. The energy dissipation increases in Fe-site doping, while it decreases in Co-site doping.

**Figure 9 nanomaterials-12-02839-f009:**
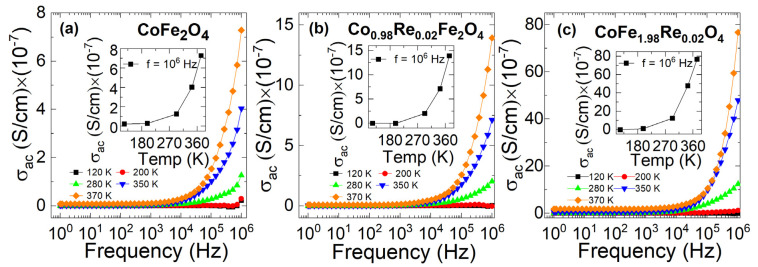
Variation in the AC conductivity of (**a**) CoFe_2_O_4_, (**b**) Co_0.98_Re_0.02_Fe_2_O_4_, and (**c**) CoFe_1.98_Re_0.02_O_4_ nanoparticles with different temperatures. The AC conductivity increases at high frequencies. The AC conductivity increases in both Fe-site doping and Co-site doping.

**Table 1 nanomaterials-12-02839-t001:** Refined structural parameters of Re-doped nanoparticles. The grain size decreases with the doping of rhenium.

Material	CoFe_2_O_4_	Co_0.98_Re_0.02_Fe_2_O_4_	Co_0.95_Re_0.05_Fe_2_O_4_	CoFe_1.98_Re_0.02_O_4_	CoFe_1.95_Re_0.05_O_4_
S. G	Fd3m	Fd3m	Fd3m	Fd3m	Fd3m
a = b = c (Å)	8.375	8.387	8.398	8.387	8.395
V (A)^3^	589.271	589.165	589.206	589.607	589.389
R_p_	0.861	0.855	0.866	0.878	0.859
R_wp_	1.188	1.177	1.187	1.178	1.129
GOF	1.025	1.014	1.027	1.018	0.975
Grain size (nm)	54.125	52.909	51.916	51.588	49.349

**Table 2 nanomaterials-12-02839-t002:** Magnetic parameters (*M_s_*, *M_r_*, *M_r_*/*M_s_*, *H_c_*, and *K*) of Re-doped nanoparticles were extracted at 300 K and 4 K. With the increased Re doping ratio, the magnetic parameters reduced in Co-site doping, while they were enhanced in Fe-site doping.

Samples	*M_s_* (emu/g)		*M_r_* (emu/g)		*M_r_*/*M_s_*		*H_c_* (*Oe*)		*K* (*erg*/*Oe*) ×10^4^		*n_B_*	
	300 K	4 K	300 K	4 K	300 K	4 K	300 K	4 K	300 K	4 K	300 K	4 K
CoFe_2_O_4_	80	80	26	68	0.32	0.85	817	4704	6.68	38.40	3.36	3.36
Co_0.98_Re_0.02_Fe_2_O_4_	79	79	19	67	0.24	0.85	680	4537	5.48	36.56	3.32	3.32
Co_0.95_Re_0.05_Fe_2_O_4_	78	77	18	66	0.23	0.86	735	4150	5.86	32.61	3.28	3.24
CoFe_1.98_Re_0.02_O_4_	80.5	81	28	73	0.35	0.90	727	7072	5.98	58.46	3.38	3.40
CoFe_1.95_Re_0.05_O_4_	81	83	27	75	0.33	0.90	712	6801	5.89	57.33	3.40	3.47

## Data Availability

The data presented in this study are available on request from the corresponding author. The data are not publicly available to make sure of their proper usage.

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
