# Peer review of "Effects of Rhenium Substitution of Co and Fe in Spinel CoFe2O4 Ferrite Nanomaterials"

_nanomaterials, 2022, doi:10.3390/nano12162839_

Round 1
Reviewer 1 Report
The article ZHeng et al. reports influence of Rhenium on CoFe2O4 nanoparticles. A comparative study of structural and magnetic properties for obtained NPs was carried out. The particles were characterized by XRD, SEM, Raman, and VSM. The authors showed that doping by Re changes structural and magnetic properties of ferrite nanoparticles. The data obtained are especially important in the field of catalysis and semi-conductors, since Re properties.
This is an interesting study deserving publication in Nanomaterials. However, there are a number of issues in the text requiring a thorough revision. First of all, in the paper there is no Materials and Apparatus section. Except some data about experiments procedure, no information about types of apparatus (Model, company, etc.) are given.
Line 54-55: Some citations should be added to the sentence.
Line 76: influence not influences
Experimental section: Authors claim that two different types of nanoparticles were obtained, where Re is substituted to A or B site of CoFe2O4 spinel structure. If so, what is the difference in the sol-gel synthesis procedures? Only in the stoichiometric ratio of ingredients? Names of the np’s suggest that the procedures should be different. Further in the text authors claim that Re occupies octahedral or tetrahedral sites, but in the experimental part there is no information, what was changed to achieve such effect.
Table 1. Some values has 2 or 3 decimal places, it should be uniformed. What are Rp and Rwp parameters? It is not explained in the text. Crystallite size calculated from Scherrer formula should be reconsidered. Giving this value with accuracy to 2 or 3 decimal places is not very accurate. This formula assumes that all crystallites are spheres, what is not seen on SEM images. Also, no measurement errors are given.
Figures 2 and 6 are difficult to read. They should be simplified.
Figure 3 – SEM images presents nanoparticles size, one example for each sample. In my opinion, to compare this parameter between each other, more nanoparticles should be measured from the images, and then compare for example size distribution. Moreover, measured values are even 20 times bigger than calculated form Scherrer formula. It is not commented in the text.
Figure 4b – In my opinion difference in wavenumbers is very small, it is not giving any important information. Also, error bars should be added.
Conclusion part is more a summary of the paper, no conclusions are given.
Author Response
Manuscript ID: nanomaterials-1841082
Type of manuscript: Article
Title: Effects of Rhenium substitution of Co and Fe in spinel CoFe2O4 ferrite
nanomaterials
Dear Sir/ Madam
Thanks for your e-mail regarding the revision of our manuscript. Now the article has
been thoroughly revised in the light of valuable reviewers' comments and suggestions.
An effort has been made to address all of the reviewers' suggestions in the revised
version of the manuscript. We are hopeful that the revised manuscript will receive
approval. The following blue words are our responses and discussion of the reviewers'
concerns.
Response to Reviewer's Queries
Reviewer # 1
This is an interesting study deserving publication in Nanomaterials. However, there are a
number of issues in the text requiring a thorough revision. First of all, in the paper, there is no
Materials and Apparatus section. Except for some data about experiments procedure, no
information about types of apparatus (Model, company, etc.) are given.
[Reply] We thank the Reviewer for her/his nice summary and kind recommendation
of our work. The details information about the types of apparatus is added in the main
text.
Line 54-55: Some citations should be added to the sentence.
[Reply] In the introduction section, we added citations in Line 54-55.
Line 76: influence not influences
[Reply] We have corrected the spelling mistakes as mentioned by the reviewer.
Experimental section: Authors claim that two different types of nanoparticles were obtained,
where Re is substituted to A or B site of CoFe2O4 spinel structure. If so, what is the difference in
the sol-gel synthesis procedures? Only in the stoichiometric ratio of ingredients? Names of the
np's suggest that the procedures should be different. Further in the text authors claim that Re
occupies octahedral or tetrahedral sites, but in the experimental part there is no information,
what was changed to achieve such effect.[Reply] The same method was adopted to synthesize these nanoparticles of
Co1-xRexFe2O4 and CoFe2-xRexO4 (0 ≤ x ≤ 0.05). Yes, the stoichiometric ratio of
ingredients is replaced. The explanation and the synthesized process were revised in
the experimental section. In this study, we were interested in investigating the effects
of Rhenium substitution for Co and Fe on the structural and physical properties of
inverse-spinel CoFe2O4.
Table 1. Some values has 2 or 3 decimal places, it should be uniformed. What are Rp and
Rwp parameters? It is not explained in the text. Crystallite size calculated from Scherrer
formula should be reconsidered. Giving this value with accuracy to 2 or 3 decimal places is not
very accurate. This formula assumes that all crystallites are spheres, what is not seen on SEM
images. Also, no measurement errors are given.
[Reply] The text about the R factor, such as profile factor (Rp), weighted Profile factor
(Rwp), and goodness factor obtained from Rietveld analysis and calculated crystallite
size from Scherrer formula were tabulated in the revised manuscript. Recently
Abdullah Kepcetoglu et al [doi:10.3906/kim-2008-59] synthesized the Rhenium
nanoparticles with sphere shape and have been used in several applications such as
tumor treating therapies and coating (plastics, metals, textiles) technologies. However,
the slight variation in lattice parameters are most probably due to stresses and strains
produced by the presence of these nanoparticles over inter-granular spaces in bulk,
causing to change in the shape of rhenium nanoparticles. Now the errors bar was
added to the revised manuscript.
Figures 2 and 6 are difficult to read. They should be simplified.
[Reply] In the revised manuscript, Figures 2 and 6 were simplified.
Figure 3 – SEM images presents nanoparticles size, one example for each sample. In my
opinion, to compare this parameter between each other, more nanoparticles should be
measured from the images, and then compare for example size distribution. Moreover,
measured values are even 20 times bigger than calculated form Scherrer formula. It is not
commented in the text.
[Reply] We thank the Reviewer again for her/his suggestion, The SEM images
presentation nanoparticles size and text are thoroughly revised in the manuscript.Figure 4b – In my opinion difference in wavenumbers is very small, it is not giving any important
information. Also, error bars should be added.
[Reply] According to the referee's opinion in figure 4b, error bars were added.
Conclusion part is more a summary of the paper, no conclusions are given.
[Reply] The conclusion text was revised in the new version of the manuscript.

Reviewer 2 Report
The reported spinel cobalt ferrite by Wang et al has special importance because it is hard due to its high coercivity, high chemical stability, and high mechanical hardness.
Doping with rare earth (RE) in cobalt ferrites caused a rising trend of the nanomagnetism phenomenon as well as modification of their properties. The properties of these ferrites are dependent on the nature of the cations, their charges, and their distribution between the tetrahedral (A) and the octahedral (B) sublattices of the spinel structure. Therefore, Yb, Ce, and Sm have been studied as dopants in the cobalt ferrite and the reports concluded that electrical conductivity decreases with increasing dopants.
The authors have reported the very rarest element Rhenium in their work, which is not widely known. Although Sol-gel synthesis is well known the obtained particle size is feasible, and the samples were ferromagnetic.
In this reviewer’s opinion, the submitted work to Nanomaterials MDPI is a suitable journal and the article is well presented, and well written with appropriate physical and conductivity properties. However, some revision is required before rendering a final decision. It requires some clarity in some areas.
My specific comments are below.
· The rationale for using Rhenium as a substitution needs to be well addressed. Why Re? What initiated the authors to choose the rarest earth element Re?
· Rare earth substituted ferrite nanoparticles were synthesized and published by various methods such as the coprecipitation method, the combustion techniques, the microemulsion method, the forced hydrolysis in polyol, and the sonochemical method. Give a brief view on these methods and introduce sol-gel and its importance in the section introduction. This will enable the reader to understand the synthesis.
· Are the ionic radii of Rhenium and Fe3+ identical?
· Page 6, line 174: why the agglomeration is severely reduced by adding Re to the Fe-site?
· What is the conclusion that has been derived from Raman's analysis?
· The dielectric behavior and AC conductivity results with respect to their hopping of charges and behavior need to be compared to similar reports (such as ChemPlusChem 9 (81) 964; Electrochim Acta 137 (2014) 497) found in the literature.
· The magnetic properties of Re-doped cobalt ferrite strongly depend on the size and shape of the nanoparticles which are closely related to the images shown in SEM. Therefore, correlating the discussion with the physical properties of the sample is essential.
· How do the total magnetic moments compare to that of other dopants reported in the literature for ferrites?
Author Response
Manuscript ID: nanomaterials-1841082
Type of manuscript: Article
Title: Effects of Rhenium substitution of Co and Fe in spinel CoFe2O4 ferrite
nanomaterials
Dear Sir/ Madam
Thanks for your e-mail regarding the revision of our manuscript. Now the article has
been thoroughly revised in the light of valuable reviewers' comments and suggestions.
An effort has been made to address all of the reviewers' suggestions in the revised
version of the manuscript. We are hopeful that the revised manuscript will receive
approval. The following blue words are our responses and discussion of the reviewers'
concerns.
Reviewer # 2
The reported spinel cobalt ferrite by Wang et al has special importance because it is hard due to
its high coercivity, high chemical stability, and high mechanical hardness.
Doping with rare earth (RE) in cobalt ferrites caused a rising trend of the nanomagnetism
phenomenon as well as modification of their properties. The properties of these ferrites are
dependent on the nature of the cations, their charges, and their distribution between the
tetrahedral (A) and the octahedral (B) sublattices of the spinel structure. Therefore, Yb, Ce, and
Sm have been studied as dopants in the cobalt ferrite and the reports concluded that electrical
conductivity decreases with increasing dopants.
The authors have reported the very rarest element Rhenium in their work, which is not widely
known. Although Sol-gel synthesis is well known the obtained particle size is feasible, and the
samples were ferromagnetic.
In this reviewer's opinion, the submitted work to Nanomaterials MDPI is a suitable journal, and
the article is well presented, and well written with appropriate physical and conductivity
properties. However, some revision is required before rendering a final decision. It requires
some clarity in some areas.
My specific comments are below[Reply] We thank the Reviewer for the comprehensive summary and the positive
comment of our work.
The rationale for using Rhenium as a substitution needs to be well addressed. Why Re? What
initiated the authors to choose the rarest earth element Re?
[Reply] The controlled substitution and search for new dopants have recently drawn
intensive research attention to improving the cobalt ferrite's structure and physical
properties. Rhenium belongs to the 5d3 transition metal, which has been reported as a
superior dopant of some semiconductors and oxides to change the properties of the
hosting materials. Teli et al. discussed the potential spintronic and optoelectronic
application of Rhenium-doped alkaline earth oxides through density functional theory
(DFT) calculation [1]. Assadi et al. recently predicted through a theoretical
calculation that ReFe2O4 occupies an unconventional ferrimagnetic state with
potential application in spintronics [2]. Moreover, Heidari et al. reported rhenium
nanoparticles' superior thermal plasmonic characteristics and their potential
application in tumor treatment [3]. However, such efforts have been explored in
studies of even broader physical phenomena. In this regard, we synthesized the
Rhenium (Re) doped cobalt ferrite nanoparticles.
Rare earth substituted ferrite nanoparticles were synthesized and published by various
methods such as the coprecipitation method, the combustion techniques, the microemulsion
method, the forced hydrolysis in polyol, and the sonochemical method. Give a brief view on
these methods and introduce sol-gel and its importance in the section introduction. This will
enable the reader to understand the synthesis.
[Reply] The brief view and importance of the sol-gel method already mentioned in the
introduction section, Page 2, lines 56-59.
Are the ionic radii of Rhenium and Fe3+ identical?
[Reply] Recently, Assadi et al. [2] Reported that the ionic radius of Fe can explain the
expansion of the volume upon Re substitution at the tetrahedral site. The radius of
high-spin Fe2+ in tetrahedral site of ReFe2O4 is 0.63 Å. In magnetite, the tetrahedral
site is occupied by Fe3+ with a smaller radius of 0.49 Å. The Re4+ radius in octahedral coordination is 0.63 Å which is quite close to the replaced Fe3+ radius (0.65 Å) and is
not expected to be a substantial drive in the structural transformation.
Page 6, line 174: why the agglomeration is severely reduced by adding Re to the Fe-site?
[Reply] The agglomeration of nanoparticles can be due to the interactions of the
magnetic surface of nanoparticles. Firstly, the inhomogeneous strain distribution can
be created due to the difference in ionic radii of the cations so that inhomogeneous
stresses and strains produced by the presence of these nanoparticles over
inter-granular spaces in bulk can form the particles with a random size and irregular
shape [4]. Thus, synthesized nanoparticles' morphology, uniformity, and
agglomeration can be changed in the presence of doping elements, either we dopped
Co-sites or Fe-sites. Secondly, the Re4+ radius in octahedral coordination is 0.63 Å
which is quite close to the replaced Fe3+ radius (0.65 Å) [2].
What is the conclusion that has been derived from Raman's analysis?
[Reply] Raman is a useful tool to detect the structural change, especially the variation
of bonds between atoms. In our result, we discovered with an increased doping ratio
that most of the peak positions that correspond to movement of the tetrahedral
sublattice shift to a higher energy side, and we find that the doping of Re induces the
distortion of the crystal structure. Moreover, a structural distortion on the octahedral
lattice is also observed for iron site doping samples. Further clarification has been
added to the conclusion section in the revised manuscript.
The dielectric behavior and AC conductivity results with respect to their hopping of charges
and behavior need to be compared to similar reports (such as ChemPlusChem 9 (81) 964;
Electrochim Acta 137 (2014) 497) found in the literature.
[Reply] Comparison of the dielectric and AC conductivity behaviour with similar
spinel ferrite nanoparticles has already been added to the revised manuscript.
The magnetic properties of Re-doped cobalt ferrite strongly depend on the size and shape of
the nanoparticles which are closely related to the images shown in SEM. Therefore, correlating
the discussion with the physical properties of the sample is essential.
[Reply] We agree with the reviewer that the size and shape of the nanoparticles
determine the magnetic properties like saturation magnetization and coercivity. SEM images show particles agglomerate with an increase of Re to Co, while reduced
agglomeration is observed for Fe-site doping. The surface-to-volume ratio of the
particle and the orientation of the magnetic moment that can be changed by clustering
influence the saturation magnetization, resulting in a decrease of Ms in Co-site doping
and an increase of Ms in Fe-site doping. The coercivity are influenced by structural
anisotropy. A detailed discussion has been included in the revised manuscript.
How do the total magnetic moments compare to that of other dopants reported in the literature
for ferrites?
[Reply] We calculated the magnetic moments according to the formula represented in
Ghorbani et al’s work [5]. The magnetic moment increases with the decrease of Co-site
doping ratio, which the change of magnetic anisotropy can explain. The magnetic
anisotropy decreases for Co-site doping, leading to a more random orientation of
moment and a decrease in net magnetic moment. Similarly, the anisotropy of Fe-site
doping increases, resulting in an increase in the total magnetic moment with an increase
in doping ratio. Compared with other dopants, Re-doped cobalt ferrite maintains a
higher total magnetic moment [5, 6], indicating an increase of anisotropy result from
doping Re.
References
[1] Teli, N. A.; Sirajuddeen, M. M. S. First-principles calculations of the electronic,
magnetic and optical properties of rhenium-doped alkaline earth oxides. Phys. Scr.
2019, 95, 025801.
[2] Assadi, M. H. N.; Fronzi, M.; Hanaor, D. A. H. Unusual ferrimagnetic ground
state in rhenium ferrite. Eur. Phys. J. Plus 2021, 137, 21.
[3] Heidari, A.; Schmitt, K.; Henderson, M.; Besana, E. Orientation Rhenium
nanoparticles delivery target on human gum cancer cells, tissues and tumors under
synchrotron radiation. J. dent. sci. oral maxillofac. res. 2019, 5, 1-18.
[4] Sharma, R.; Thakur, P.; Sharma, P.; Sharma, V. Ferrimagnetic Ni2+ doped Mg-Zn
spinel ferrite nanoparticles for high density information storage. J. Alloys Compd.
2017, 704, 7–17.[5] Ghorbani, H.; Eshraghi, M.; Dodaran, A. S. Structural and magnetic properties of
cobalt ferrite nanoparticles doped with cadmium. Physica B Condens. Matter 2022,
634, 413816.
[6] Islam, M.; Khan, M. K. R.; Kumar, A.; Rahman, M. M.; Abdullah-Al-Mamun, M.;
Rashid, R.; Haque, M. M.; Sarker, M. S. I., Sol–gel route for the synthesis of CoFe2–
xErxO4 nanocrystalline ferrites and the investigation of structural and magnetic
properties for magnetic device applications. Acs Omega 2022, 7, 20731-20740.

Reviewer 3 Report
The article Effects of Rhenium substitution of Co and Fe in spinel CoFe2O4 ferrite nanomaterials is devoted to the study of the properties of magnetic nanoparticles doped with Rhenium under the condition of substitution of cobalt or iron. In general, this article has sufficient novelty and practical significance in the field of nanotechnology and methods for obtaining magnetic nanoparticles. The authors used a fairly large number of methods to assess the structural and magnetic parameters of the objects under study. The article has a high level of novelty and can be accepted for publication after the authors answer a number of questions that arose while reading it.
1. The results of X-ray diffraction require a significant improvement in presentation, the authors should present comparative diffraction patterns of the change in the position of the main peaks depending on the concentration of Rhenium in the composition, and also describe the effect of increasing the parameters in comparison with the ionic radii of the elements in the structure.
2. The accuracy in determining the dimensions using the Scherrer formula is too high, this formula allows you to determine the average size of crystallites, such accuracy in the indicated values ​​is not needed, and measurement errors are not given either.
3. The authors need to give a correlation between the size data determined by X-ray diffraction and scanning electron microscopy.
4. The authors should give a more detailed description of the behavior of the magnetic properties depending on the concentration of Rhenium.
5. In the abstract, it is necessary to indicate in more detail not only the main results obtained, but also the purpose and relevance in the selected research objects, to emphasize the Rhenium substitution effect.
Author Response
Manuscript ID: nanomaterials-1841082
Type of manuscript: Article
Title: Effects of Rhenium substitution of Co and Fe in spinel CoFe2O4 ferrite
nanomaterials
Dear Sir/ Madam
Thanks for your e-mail regarding the revision of our manuscript. Now the article has
been thoroughly revised in the light of valuable reviewers' comments and suggestions.
An effort has been made to address all of the reviewers' suggestions in the revised
version of the manuscript. We are hopeful that the revised manuscript will receive
approval. The following blue words are our responses and discussion of the reviewers'
concerns.
Response to Reviewer's Queries
Reviewer # 3
The article Effects of Rhenium substitution of Co and Fe in spinel CoFe2O4 ferrite
nanomaterials is devoted to the study of the properties of magnetic nanoparticles doped with
Rhenium under the condition of substitution of cobalt or iron. In general, this article has
sufficient novelty and practical significance in the field of nanotechnology and methods for
obtaining magnetic nanoparticles. The authors used a fairly large number of methods to assess
the structural and magnetic parameters of the objects under study. The article has a high level
of novelty and can be accepted for publication after the authors answer a number of questions
that arose while reading it.
- The results of X-ray diffraction require a significant improvement in presentation, the authors
should present comparative diffraction patterns of the change in the position of the main peaks
depending on the concentration of Rhenium in the composition, and also describe the effect of
increasing the parameters in comparison with the ionic radii of the elements in the structure.
[Reply] We thank the Reviewer for the overall positive comments on this work as
well as the valuable suggestions. The results of the X-ray diffraction pattern were combined, and increasing parameters were explained with the ionic radii of the
elements in the revised manuscript.
- The accuracy in determining the dimensions using the Scherrer formula is too high, this
formula allows you to determine the average size of crystallites, such accuracy in the indicated
values is not needed, and measurement errors are not given either.
[Reply] The average size of crystallites was recalculated, and the errors bar was added in the
revised manuscript.
- The authors need to give a correlation between the size data determined by X-ray diffraction
and scanning electron microscopy.
[Reply] The SEM and XRD graphs were improved, and a brief explanation about the
size data was added.
- The authors should give a more detailed description of the behavior of the magnetic
properties depending on the concentration of Rhenium.
[Reply] The detailed description of magnetic properties depending on the
concentration of Rhenium was added in the revised manuscript.
- In the abstract, it is necessary to indicate in more detail not only the main results obtained,
but also the purpose and relevance of the selected research objects, to emphasize the
Rhenium substitution effect.
[Reply] The abstract was revised in the manuscript.

Round 2
Reviewer 1 Report
Thank you for all the answers.
Reviewer 2 Report
In this reviewer's opinion, the revised part of the manuscript is good and the authors have addressed my queries reasonably well.
Reviewer 3 Report
The authors answered all the questions, the article can be accepted for publication.